# Slow magnetic relaxation in a europium(II) complex

Dylan Errulat[1,6], Katie L. M. Harriman[1,6], Diogo A. Gálico [1], Elvin V. Salerno[2], Johan van Tol [2], Akseli Mansikkamäki [3], Mathieu Rouzières [4], Stephen Hill [2,5] ✉, Rodolphe Clérac [4] ✉ & Muralee Murugesu [1] ✉

Single-ion anisotropy is vital for the observation of Single-Molecule Magnet (SMM) properties (i.e., a slow dynamics of the magnetization) in lanthanide-based systems. In the case of europium, the occurrence of this phenomenon has been inhibited by the spin and orbital quantum numbers that give way to $J = 0$ in the trivalent state and the half-filled population of the 4f orbitals in the divalent state. Herein, by optimizing the local crystal field of a quasi-linear bis(silylamido) $Eu^{II}$ complex, the $[Eu^{II}(N\{SiMePh_2\}_2)_2]$ SMM is described, providing an example of a europium complex exhibiting slow relaxation of its magnetization. This behavior is dominated by a thermally activated (Orbach-like) mechanism, with an effective energy barrier of approximately 8 K, determined by bulk magnetometry and electron paramagnetic resonance. Ab initio calculations confirm second-order spin-orbit coupling effects lead to non-negligible axial magnetic anisotropy, splitting the ground state multiplet into four Kramers doublets, thereby allowing for the observation of an Orbach-like relaxation at low temperatures.

Single-molecule magnets (SMMs) have garnered much attention owing to their potential application in molecular-level memory storage devices as well as for quantum information technologies[1–4]. These SMM properties arise from an intrinsic energy barrier to the reorientation of the magnetic moment. The strong unquenched orbital contribution of the core 4f-orbitals leads to large inherent magnetic anisotropy in many of the lanthanide series, and the half-integer angular momentum projection of ions such as $Dy^{III}$ ($^6H_{15/2}$) ensures a doubly degenerate ground state, as dictated by Kramers theorem[5]. Recently, the use of lanthanide (Ln) ions in the design of SMMs has led to significant breakthroughs in performance and operation temperatures beyond that of liquid nitrogen ($T = 77$ K)[6–9]. These remarkable improvements are largely thanks to the exceptional synthetic control of the crystal field splitting and axiality (i.e., ligand design and coordination geometry)[10].

Beyond the crystal field, the inherent magnetic moment of the metal center is an important consideration, as a greater magnitude of $J$ can reduce ground-state quantum tunneling of the magnetization (QTM) between degenerate states and lead to overall improvements in the coercivity and blocking temperature[11,12]. However, altering the effective magnetic moment of a given 4f ion is challenging as the trivalent oxidation state is the most stable for Ln ions[13]. New discoveries in synthetic 4f chemistry have resulted in the expansion of accessible oxidation states[14,15]. These discoveries have led to record magnetic moments for the divalent complexes $[(Cp')_3Ln^{II}]^-$ (Cp' = trimethylsilylcyclopentadienyl, Ln = $Dy^{II}$, $Ho^{II}$), originating from the $4f^n5d^1$ configuration[16]. Despite this attractive feature, $[(Cp')_3Ln^{II}]^-$ do not display slow magnetization dynamics. Nevertheless, ab initio calculations have recently demonstrated that divalent complexes have the potential to possess very strong magnetic anisotropy and very large

[1]Department of Chemistry and Biomolecular Sciences, University of Ottawa, Ottawa Ontario K1N 6N5, Canada. [2]National High Magnetic Field Laboratory, Florida State University, Tallahassee, FL 32310, USA. [3]NMR Research Unit, University of Oulu, P. O. Box 3000, 90014 Oulu, Finland. [4]Univ. Bordeaux, CNRS, CRPP, UMR 5031, F-33600 Pessac, France. [5]Department of Physics, Florida State University, Tallahassee, FL 32306, USA. [6]These authors contributed equally: Dylan Errulat, Katie L. M. Harriman. ✉e-mail: shill@magnet.fsu.edu; rodolphe.clerac@u-bordeaux.fr; m.murugesu@uottawa.ca

magnetization blocking barriers ($\Delta/k_B > 3000$ K) when the geometry around the metal ion is highly symmetric and/or low coordinate[17]. As an illustration, the divalent metallocene [Tb$^{II}$(Cp$^{iPr5}$)$_2$] (Cp$^{iPr5}$ = pentai-sopropylcyclopentadienyl, Ln = Tb$^{II}$) possesses an impressive barrier of 1738 K (1205 cm$^{-1}$) and a blocking temperature of 52 K[18]. Ultimately, a clear trend has yet to be established for the effects of oxidation states on the splitting of the ground and excited states, as well as variations in metal-ligand covalency[19,20].

The spin contribution to the magnetic moment is the highest in the case of a half-filled 4f-shell (4f$^7$), such as Gd$^{III}$. However, this electronic configuration is largely considered to be magnetically isotropic as a result of the spherically symmetric charge density of the $^8S_{7/2}$ ground state that lacks orbital angular momentum ($S = 7/2$, $L = 0$). Nonetheless, some Gd$^{III}$ complexes displaying slow magnetization relaxation have been documented[21–35]. Combined with the inclusion of the isoelectronic Tb$^{IV}$, the 4f$^7$ valence series (Eu$^{II}$, Gd$^{III}$, and Tb$^{IV}$) provides unique insight into the minutia that govern the electronic properties of lanthanides beyond the trivalent oxidation state, serving as an excellent model to examine the precise effects of the coordination environment on the electronic structures of Ln ions with uncommon oxidation states. However, until now, the magnetic studies of Eu-based complexes have been limited to their static susceptibility properties. In order to expand the prevalence of elements displaying slow magnetization relaxation, we present herein the quasi-linear complex, [Eu$^{II}${N(SiMePh$_2$)$_2$}$_2$] (**1**), which is to the best of our knowledge, the first example of a europium SMM.

## Results and discussions
### Synthesis and structural analysis
Compound **1** was prepared by the reaction of one equivalent of EuI$_2$ with two equivalents of K(THF)N(SiMePh$_2$)$_2$ in THF, in accordance with the previously reported procedure for [Yb$^{II}${N(SiMePh$_2$)$_2$}$_2$] (Fig. 1a)[36]. Crystallization from an *n*-hexane solution gave [Eu$^{II}${N(SiMePh$_2$)$_2$}$_2$] (**1**) as orange crystals in ~68% yield.

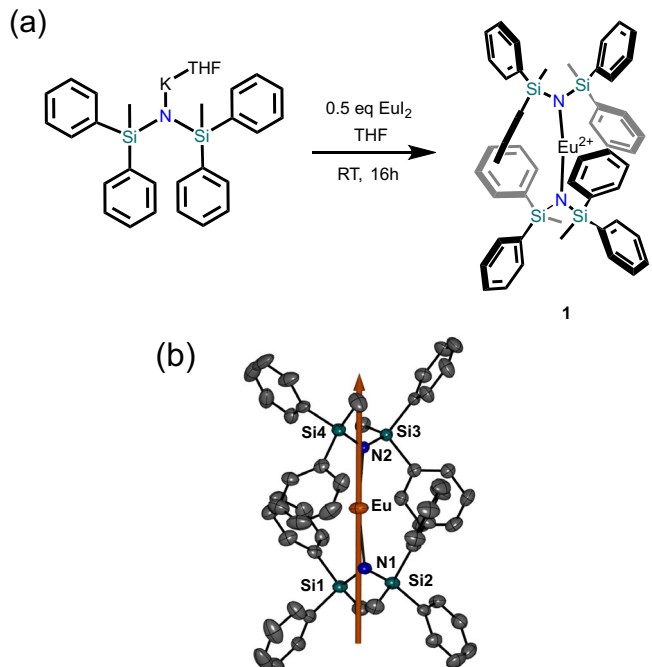

**Fig. 1 | Synthesis and structure of Eu{N(SiMePh$_2$)$_2$}$_2$ (1). a** Reaction scheme for the synthesis of **1**. **b** ORTEP molecular model with 50% thermal probability ellipsoids. H-atoms have been omitted for clarity. Orange, europium; teal, silicon; blue, nitrogen; gray, carbon. The solid orange line represents the orientation of the principal anisotropy axis of the zero-field split $^8S_{7/2}$ multiplet.

Single-crystal X-ray diffraction reveals **1** crystallizes in the monoclinic $P2_1/c$ space group (Supplementary Table 1). The complex has a near-linear geometry with an N−Eu−N angle of 169.808(83)° and metal-nitrogen distances of 2.4884(26) Å and 2.4612(26) Å, comparable to the Yb$^{II}$ analog[36]. The slightly longer bond lengths in **1** compared to the Yb$^{II}$ analogue is consistent with the increased ionic radius of the Eu$^{II}$ ion. Two phenyl moieties from each ⁻N(SiMePh$_2$)$_2$ ligand act to encapsulate the metal ion with Eu•••C interactions between 2.93 and 3.04 Å, preventing any further aggregation or interactions with solvent molecules. The individual complexes are well separated in the lattice by a distance of 10.64 Å (Supplementary Fig. 1).

### Photoluminescence spectroscopy
The excitation spectrum of **1** obtained at 10 K (Supplementary Fig. 2) reveals a broad signal in the UV range at 270 nm due to the ligand and Eu$^{II}$ 5d centered absorptions. A single component centred at 518 nm (19,305 cm$^{-1}$) is observed in the emission spectrum (Supplementary Fig. 3a), characteristic of Eu$^{II}$ 5d → 4f transitions. A blue shift of the transition is observed at 293 K, with the center being shifted to 496 nm (20,161 cm$^{-1}$). At room temperature, the emitter level lifetime is 2.45 μs (Supplementary Fig. 4). It is noteworthy that vibrational components are observable at room temperature, suggesting strong vibronic coupling. Deconvolution of the emission band results in seven different components equally separated (-1550 cm$^{-1}$), suggesting possible vibronic coupling between the excited 5d state and the vibration located at 1561 cm$^{-1}$ in the FT-IR (Supplementary Fig. 3b). Attempts to oxidize the divalent complex **1** using silver salts, particularly AgAl{OC(CF$_3$)$_3$}$_4$, resulted in the formation of a silver mirror (Ag$^I$ → Ag$^0$) and strong red emission visible to the naked eye when excited under a 405 nm UV laser. At first glance, this result might suggest the successful oxidation of **1** to the trivalent oxidation state with the characteristic red emission of Eu$^{III}$. While single crystals of the oxidized product could not be isolated, the excitation spectrum of the obtained crude oil was collected in C$_6$H$_5$F (Supplementary Fig. 5). The characteristic broad emission of the Eu$^{II}$ 5d → 4f transitions was observed with the band red-shifted to 756 nm (13,227 cm$^{-1}$). This drastic red shift indicates a significant change in the metal's coordination environment, suggesting that a ligand-centred oxidation is taking place in the presence of Ag$^I$. As the Eu$^{II/III}$ redox potential (−0.35 V *vs.* SCE) is the most positive of the lanthanides, the divalent oxidation state is particularly stable[37]. In fact, the reaction of K(THF)N(SiPh$_2$Me)$_2$ with AgAl{OC(CF$_3$)$_3$}$_4$ in the absence of oxidizable Eu$^{II}$ still resulted qualitatively in the formation of Ag$^0$ and an intractable product whose structure could not be determined. This echoes several of the challenges observed in previous reports by Mills and co-workers when attempting to oxidize bis(silylamido) Eu$^{II}$ complexes, where the bis(silylamide) ligands undergo non-reversible oxidation in place of the metal center[38,39].

### Computational studies
The electronic structure of **1** was studied by computational methods (see Methods section), confirming a ground electronic configuration of 4f$^7$, which would be expected to give rise to an isotropic $^8S_{7/2}$ multiplet[40]. However, linear coordination geometries have recently been shown to stabilize states with multiple open shell configurations in coordination complexes of divalent lanthanide ions[18]. Therefore, this possibility was also considered for **1**. At the hybrid DFT level using the PBE0 exchange-correlation functional, all seven unpaired electrons are found to reside in orbitals with more than 88% 4f character. Furthermore, ab initio multireference calculations were performed[41,42], demonstrating that the $^8S_{7/2}$ ground spin multiplet is weakly split under second-order spin-orbit coupling (SOC) effects into four Kramers doublets spanning an energy range of 3−4 cm$^{-1}$ (4−6 K). The calculated *g* tensor of the $^8S_{7/2}$ multiplet is almost isotropic with principal values of 1.9975, which only differ in the fifth decimal. The

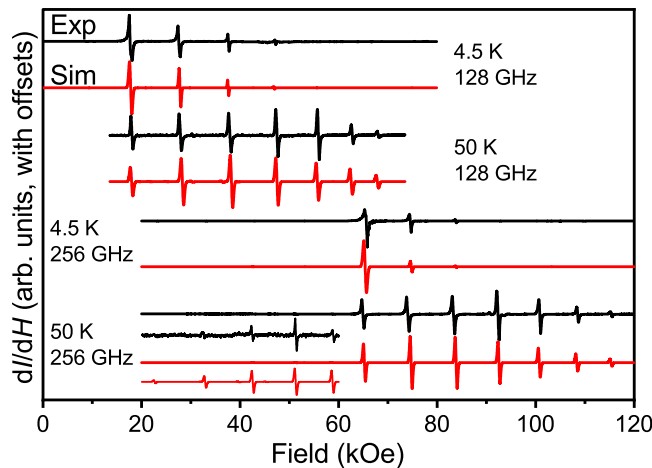

**Fig. 2 | CW high-field EPR spectra of Eu{N(SiMePh₂)₂}₂ (1).** Unconstrained powder sample recorded in derivative mode (d$I$/d$H$, where $I$ is the microwave intensity transmitted through the sample and $H$ the applied field) at 4.5 K & 128 GHz (top), 50 K & 128 GHz (second from top), 4.5 K & 256 GHz (second from bottom), and 50 K & 256 GHz (bottom). In addition, the optimal simulations (depicted in red) are presented, generated following the procedure outlined in the main text. The spectra are offset for clarity. The inset shows an expanded view of the low-field region of the 256 GHz, 50 K spectrum, highlighting weakly allowed $\Delta m_S = \pm 2$ transitions that are also captured by the simulations. Source data are provided as a Source Data file.

second-order zero-field splitting (ZFS) parameters are calculated to be $D/k_B = -0.40$ K and $E/k_B = +0.008$ K, indicating the presence of axial anisotropy. The higher-order ZFS parameters are negligible. The principal axis of the ZFS tensor is oriented along the molecular N–Eu–N pseudo axis (Fig. 1b), implying that **1** possesses non-negligible magnetic axiality, owing to the quasi linear environment provided by the bis(silylamide) ligands. It can therefore be anticipated that the ZFS will result in slow relaxation of magnetization via an Orbach-like mechanism at low temperatures.

## High-field EPR measurements

Due to the inferred magnetic anisotropy of **1**, temperature-dependent continuous-wave (CW) high-field electron paramagnetic resonance (EPR) measurements were performed on an unconstrained poly-crystalline sample, at frequencies of 128 and 256 GHz (Fig. 2)[43,44]. Seven dominant resonances are observed in the spectra recorded at each frequency, corresponding to the 2$S$ allowed EPR ($\Delta m_S = \pm 1$) transitions within the eight (= 2$S$ + 1) spin projection states associated with the $S = 7/2$ Eu$^{II}$ ion (Fig. 2); note that weakly allowed $\Delta m_S = \pm 2$ transitions are also observed in the half-field region of the spectrum (Fig. 2, inset). Excellent spectral simulations are obtained with the following ZFS parameters: $g_z = 1.998(5)$, $D/k_B = -0.659(4)$ K ($D/hc = -0.458(3)$ cm$^{-1}$), in good agreement with the above calculations and the magnetization data discussed below. In order to reproduce the observed intensity patterns, $D$-strain, $\sigma_D = 0.004$ cm$^{-1}$, and a peak-to-peak linewidth, $\sigma_{lw} = 15$ mT, were included in the 4.5 K simulations ($\sigma_D = 0.001$ cm$^{-1}$ and $\Delta_{lw} = 45$ mT at 50 K). These simulations assume alignment of the crystallites within the powder sample, which is to be expected due to the magnetic torque on the anisotropic Eu$^{II}$ species. However, the glide plane in the unit cell of **1** gives rise to symmetry equivalent molecules that are not collinear (Supplementary Fig. 1a). Consequently, the field aligns along the average anisotropy direction of the bulk crystals, which intersects the principal magnetic axes of the two sites[45]. The most accurate simulations were achieved when the angle ($\theta$) between the applied field and the principal axes of the ZFS ($D$) tensors of the molecules in the unit cell was set to 16.6(2)°. Because of this alignment, only the axial spin Hamiltonian parameters, $g_z$ and $D$, could reliably be

determined from the simulations. Due to the reactivity of **1**, several attempts to constrain the sample to prevent torquing proved unsuccessful. As such, the bulk crystals were left unconstrained in order to avoid potential decomposition.

The obtained $D$ parameter gives a magnetization reversal barrier, $\Delta_{EPR}/k_B = (S^2 - \frac{1}{4})|D|/k_B = 7.9$ K, which is in almost perfect agreement with the value determined from the ac susceptibility studies discussed below ($\Delta_{ac}/k_B = 8.2$ K, vide infra). Finally, we note that the simulated peak positions enable an unambiguous determination of the magnitude of $D$. However, in order to determine its sign, one must compare temperature-dependent spectra. As can be seen from the spectra in Fig. 2, spectral weight shifts from high- to low-field upon cooling, which is fully consistent with a negative $D$ parameter[46].

## Magnetic measurements

To probe the magnetic properties, direct current (dc) magnetic susceptibility measurements were performed on **1**. At room temperature, a $\chi T$ product of 7.8 cm$^3$ K mol$^{-1}$ is observed (Supplementary Fig. 6), which is in good agreement with the $S = 7/2$ ground state expected for an isolated Eu$^{II}$ ion with a 4f$^7$ ground state configuration (7.875 cm$^3$ K mol$^{-1}$). The $\chi T$ product remains constant down to 50 K, whereafter it decreases slightly to reach 6.3 cm$^3$ K mol$^{-1}$ at 1.85 K and 1000 Oe. This low-temperature behavior is likely the result of ZFS, as predicted by the computational results and confirmed by EPR (vide supra), and has been observed in related linear Eu$^{II}$ complexes[47]. Magnetization as a function of applied magnetic field (between 0 and 70 kOe; Supplementary Fig. 7) in the temperature range of 1.85–8 K exhibits a high-field linear variation and non-complete saturation at 70 kOe, indicative of magnetic anisotropy. A near saturation close to 7.0 $\mu_B$ at 1.85 K and 70 kOe is in good agreement with the $S = 7/2$ ground state. Non-superposition of the isothermal lines of the reduced magnetization plot (Supplementary Fig. 7) further suggests the presence of non-negligible ZFS. An attempt was made to fit the experimental $M$ vs. $H/T$ and $\chi T$ vs. $T$ data to the simplest $S = 7/2$ model with the $\hat{H} = D\hat{S}_{T,z}^2$ Hamiltonian (Supplementary Fig. 8). The experimental data are qualitatively reproduced with $D/k_B = -0.58(5)$ K ($D/hc = -0.40(7)$ cm$^{-1}$), in agreement with EPR data and computational results.

Alternating current (ac) susceptibility measurements were performed on **1**. In the absence of an applied magnetic field, the ac susceptibility does not exhibit any sign of slow relaxation of the magnetization down to 1.9 K and up to 10 kHz. Upon application of a static dc field, a clear signal in the out-of-phase component is observed, indicating the presence of slow paramagnetic relaxation. (Supplementary Fig. 9). The frequency dependence of the ac susceptibly data measured at different magnetic fields up to 10 kOe was fitted to the generalized Debye model (Supplementary Eqs. 1 and 2; Supplementary Figs. 9 and 10) from which the relaxation times ($\tau$) were extracted (Supplementary Fig. 11)[48]. To elucidate the contribution of the individual relaxation mechanisms on the spin dynamics of **1**, the relaxation times were fit to the sum of individual QTM ($B_1$, $B_2$), Direct ($A$), Raman ($C$, $C1$, $C2$), and Orbach-like ($\tau_0$, $\Delta$) relaxation mechanisms. These pathways are the most frequently used in the phenomenological models used to describe the relaxation dynamics of SMMs, which is expressed by the rate equations (Eqs. 1 and 2)[49–51]:

$$\tau^{-1} = \tau_{QTM}^{-1} + \tau_{Direct}^{-1} + \tau_{Raman}^{-1} + \tau_{Orbach}^{-1} \qquad (1)$$

$$\tau^{-1} = \frac{B_1}{1 + B_1 H^2} + AH^4 T + C\frac{1 + C_1 H^2}{1 + C_2 H^2} T^n + \tau_0^{-1} \exp\left(-\frac{\Delta\left(1 - \frac{H}{H_{sw}}\right)^n}{k_B T}\right) \qquad (2)$$

At fields below 1 kOe, the relaxation time increases with an $H^2$ dependence (Supplementary Fig. 11), which is consistent with the field dependence of a QTM process described by the first term in Eq. (2).

Fitting of the experimental $\tau$ between 0 and 800 Oe leads to the following parameters: $B_1 = 1.8(1)\ 10^4\,s^{-1}$ and $B_2 = 22.8(2.3)\ kOe^{-2}$ (Supplementary Fig. 11). At higher fields, the variation does not follow an $H^{-4}$ dependence that would indicate a Direct process, nor an $H^{-2}$ field-dependence characteristic of Raman relaxation. Indeed, this field variation of the relaxation time is modest, suggesting the presence of a thermally activated (Orbach-like) relaxation process (last term in Eq. (2)). To elucidate the relaxation dynamics of **1**, ac magnetic susceptibility as a function of temperature was measured under an applied dc field 800 Oe (Fig. 3 and Supplementary Fig. 12), for which the relaxation time is the longest and the effects of ground state QTM are minimized.

The obtained temperature dependence of $\tau$ can be modeled between 1.9 and 5 K considering a thermally activated (Orbach-like) relaxation process with an energy gap of 8.2(5) K (or 5.7(3) cm$^{-1}$; $\tau_0 = 3.1(6)\ 10^{-6}\,s$; Supplementary Figs. 13–15) and a contribution from a QTM relaxation process ($B_1$ and $B_2$ being fixed to the values obtained from the fit of the field dependence). This result indicates unambiguously that the thermally activated relaxation is dominant in this temperature region, which agrees with the analysis of the field dependence of the relaxation. Nonetheless, two factors suggest the presence of additional relaxation mechanisms: (i) the obtained pre-exponential factor ($\tau_0$) of 3.1(6) 10$^{-6}$s is larger than expected for an Orbach-like

relaxation regime; and (ii) the field dependence of the relaxation time above 1 kOe is complex and could not be reproduced by fitting to Eq. (2). Attempts to fit the experimental data with relaxation models including Raman and direct processes lead to overparameterization and did not improve the theory-experiment agreement above 0.1 T (Supplementary Fig. 15).

The Eu$^{II}$ complex presented herein exhibits slow relaxation of the magnetization, expanding the elements of the 4f$^7$ valence series capable of behaving as an SMM beyond Gd$^{III}$. This divalent complex belongs to a rare and synthetically challenging class of low-coordinate SMMs that continues to attract interest. An Orbach-like mechanism dominates the relaxation dynamics of **1**, with a relaxation barrier of 8.2 K. The highly axial coordination environment, combined with second order spin-orbit coupling, gives rise to a weak axial ZFS interaction that provides the origin of the paramagnetic relaxation. These results underscore the significance of crystal field effects in lanthanide ions, which allows us to harness the magnet-like behavior of the Eu$^{II}$ ion.

## Methods
### General methods
All operations were performed in an M. Braun glovebox under an atmosphere of purified dinitrogen or using high vacuum standard Schlenk techniques. Solvents were dried using a J.C. Meyer solvent system, degassed with successive freeze-pump-thaw cycles, and stored over activated 4 Å molecular sieves prior to use. EuI$_2$ was purchased from Sigma Aldrich in anhydrous form at ≥ 99.9% purity. Celite used for filtration was dried under vacuum while heating to 180 °C for 5 days. [K{N(SiMePh$_2$)$_2$}THF]$_n$ was prepared according to literature procedure[52]. [Eu{N(SiMePh$_2$)$_2$}$_2$] (**1**) was prepared based on our previously published synthetic procedure[36]. A brief description of the synthesis is detailed below. FT-IR spectra were recorded on a Nicolet Nexus 550 FT-IR spectrometer using transmission mode in the 4000-600 cm$^{-1}$ range; solid samples were prepared in an inert atmosphere and sandwiched between transparent NaCl plates. Elemental analyses were performed by Midwest Microlab.

### Synthesis of [Eu{N(SiMePh$_2$)$_2$}$_2$] (1)
A solution of [K{N(SiMePh$_2$)$_2$}THF]$_n$ (128 mg, 0.246 mmol) in THF (5 mL) was added to EuI$_2$ (50 mg, 0.123 mmol) in 5 mL of THF. The mixture was allowed to stir for 16 h, followed by the removal of the solvent under reduced pressure. The crude solid was suspended in n-hexane, followed by filtration through Celite. The solvent was then removed under reduced pressure until incipient crystallization was observed. The product is obtained as crystalline orange blocks in 68% yield (81 mg, 0.083 mmol). CHN Anal. Calcd. for C$_{52}$H$_{52}$N$_2$Si$_4$Eu: C, 64.43; H, 5.41; N, 2.89. Found: C, 62.85; H, 5.43; N, 3.01. FT-IR (neat, cm$^{-1}$): 487(w) 596(w), 627(w), 662(w), 703(s), 729(s), 775(s), 799(s), 852(w), 870(w), 967(m), 1031(w), 1060(w), 1110(m), 1153(w), 1183(w), 1251(w), 1298(w), 1330(w), 1423(w), 1477(w), 1561(w), 1581(w).

### Crystal growth and single-crystal X-ray diffraction analysis
Crystal growth of [Eu{N(SiMePh$_2$)$_2$}$_2$] was accomplished by concentration of a solution of **1** in n-hexane, from which bright orange blocks suitable for X-ray crystallographic study were obtained. X-ray diffraction data represents the most reliable datasets obtained from multiple trials. The crystal was submerged in Parabar 10312 (Paratone N) oil and mounted on a MiTeGen Microloop. The X-ray diffraction data for **1** was collected on a Bruker KAPPA APEX II single crystal diffractometer equipped with a sealed Mo tube source ($\lambda = 0.71073$ Å) APEX II CCD detector at a temperature of 200 K. The structure was solved by direct method with SHELXT[53] and refined based on F$^2$ with SHELLXL[54]. Hydrogen atom positions were calculated based on the geometry of related non-hydrogen atoms and treated as idealized contributions during the refinement.

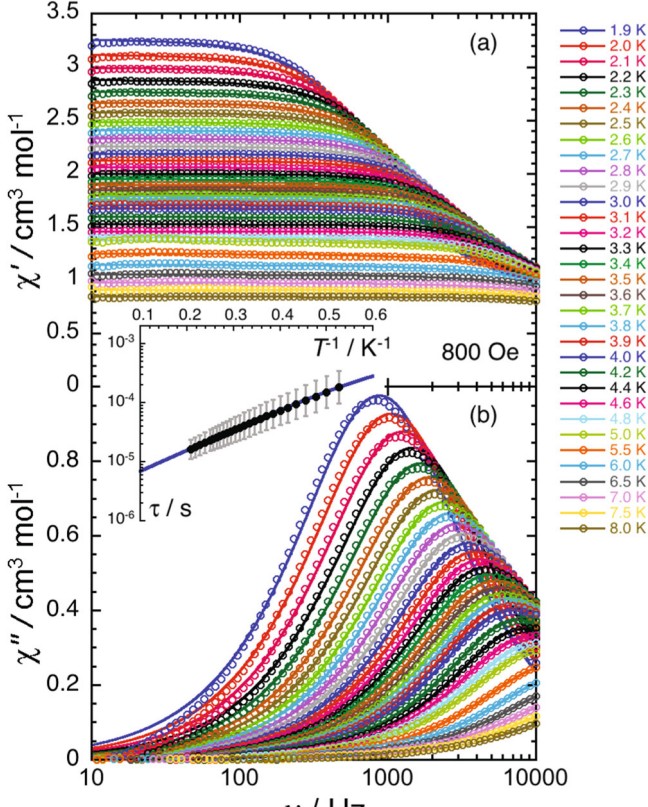

**Fig. 3 | ac magnetic susceptibility data for Eu{N(SiMePh$_2$)$_2$}$_2$ (1). a** Frequency dependence of the in-phase ($\chi'$) ac magnetic susceptibility in the range of 1.9 K – 8 K, collected at a constant applied field of 800 Oe. **b** Frequency dependence of the out-of-phase ($\chi''$) ac magnetic susceptibility from 1.9 K–8 K, collected at a constant applied field of 800 Oe. The solid lines represent the fit to the generalized Debye model at each temperature. (Inset) The average $\tau$ as a function of inverse temperature at 800 Oe, fitted from 1.9 K to 5 K. The estimated standard deviation (ESD) of the relaxation times are represented as vertical solid bars calculated from the $\alpha$-parameters of the generalized Debye fits and the log-normal distributions[87]. The blue line represents the best fit ($R^2 = 0.9998$) to a model considering both thermally activated (Orbach-like) and QTM relaxation processes (Eqs. 1 and 2). Source data are provided as a Source Data file.

## Luminescence measurements

The photoluminescence data was obtained using a QuantaMaster 8075-21 spectrofluorometer (Horiba). An ozone-free PowerArc energy 75 W xenon lamp was used as the radiation source. The excitation spectra were corrected in real-time according to the lamp intensity and the optical system of the excitation monochromator using a silicon diode as a reference. The emission spectra were corrected according to the optical system of the emission monochromator and the photomultiplier response (Hamamatsu R13456 red extended PMT). Low-temperature measurements were performed with the samples mounted inside a CS202*E-DMX-1AL closed-cycle helium cryostat system (Advanced Research Systems) controlled via a LakeShore 335 temperature controller. For the emission decay curve, we excited the sample with a 275 nm LED (Thorlabs M275L4) controlled with a LED driver (Thorlabs DC2200 Driver) and collected the emission with an Andor iStar 334 ICCD camera installed in a Shamrock SR750 spectrograph. The LED was pulsed with a 20 kHz frequency (one pulse every 50 $\mu$s) and a pulse width of 2.5 $\mu$s.

## CW EPR spectrometry

The sample was ground to a fine powder under inert atmosphere using a mortar and pestle. It was then placed in a polyethylene cup with a Teflon stopper, then shipped to the NHMFL in an ampoule which was sealed under inert atmosphere. Just prior to measurement, the ampoule was opened, and the sample quickly mounted in the spectrometer and cooled under a helium atmosphere. The transmission type spectrometer[43] employs a 17 T sweepable superconducting magnet. Microwave frequencies in the range 128–256 GHz were generated using a tunable phase-locked source followed by a series of frequency multipliers (Virginia Diodes Inc., Charlottesville, VA). The field-modulated EPR signal was generated using a wound copper coil and the signal detected using an InSb hot-electron bolometer (QMC Ltd., Cardiff UK). The final derivative-mode ($dI/dH$, where $I$ is the microwave intensity transmitted through the sample and $H$ is the applied magnetic field) EPR signal was demodulated using a lock-in amplifier at 50 kHz (Stanford Research Systems, Sunnyvale, CA). The temperature was regulated via a continuous flow helium cryostat in combination with an ITC 503 controller (Oxford Instruments, Oxford UK).

## Magnetometry

Magnetic susceptibility measurements were performed on a Quantum Design SQUID MPMS-XL magnetometer and PPMS-9 susceptometer housed at the Centre de Recherche Paul Pascal at temperatures between 1.85 and 300 K and dc magnetic fields ranging from −7 to +7 T. The ac magnetic susceptibility measurements were performed in an oscillating ac field of 1 to 6 Oe with frequencies between 0.1 Hz and 10 kHz and various dc fields (including zero). The measurements were carried out on polycrystalline samples of [Eu{N(SiMePh$_2$)$_2$}$_2$] (**1**) (16.9 and 9.8 mg) suspended in mineral oil (typically 10-12 mg) and introduced in a sealed double layered polyethylene/polypropylene bag ($3 \times 0.5 \times 0.02$ cm; 25.5 and 20.1 mg). The consistent and reproducible dc and ac measurements on these samples were combined into a single extended set. Prior to the experiments, the field-dependent magnetization was measured at 100 K to exclude the presence of bulk ferromagnetic impurities. In fact, common paramagnetic or diamagnetic materials should exhibit a perfectly linear dependence of the magnetization at 100 K that extrapolates to zero at zero dc field; the samples appeared to be free of any ferromagnetic impurities. The magnetic susceptibilities were corrected for the sample holder, the mineral oil, and the intrinsic diamagnetic contributions.

## Computational details

The geometry of **1** was extracted from the crystal structure. The positions of hydrogen atoms were optimized using density functional theory (DFT), while the positions of heavier atoms were kept frozen to their crystal-structure coordinates. The optimization was carried out using the *Amsterdam Modeling Suite* (AMS) version 2020.101[55–58]. The pure GGA exchange-correlation functional PBE[59,60] was used along with Grimme's DFT-D3 dispersion correction[61] with the Becke–Johnson damping function[62]. Scalar relativistic effects were treated using the zeroth-order regular approximation (ZORA)[63–65]. The default Slater-type basis sets designed for ZORA calculations were used in all DFT calculations[66]. A valence-polarized triple-$\zeta$ quality basis (TZP) was used for the Eu ion with all orbitals up to 5$p$ treated as a frozen core. Valence-polarized double-$\zeta$ quality basis sets were used for the remaining atoms. The 1$s$ electrons in C and N atoms and all electrons up to 2$p$ in Si were treated as frozen cores. In order to study the electron configuration, a final single-point DFT calculation was carried out on 1 using the hybrid PBE0 exchange-correlation functional and all-electron triple-$\zeta$ quality basis sets with two sets of polarization functions (TZ2P) for all atoms[67,68].

The electronic structure of **1** was further studied by multi-reference ab initio calculations. These were conducted using the OpenMolcas code version 21.02[69,70]. Two state-averaged complete active space self-consistent field (SA-CASSCF) calculations were carried out[71–75]. The active space consisted of the seven 4f orbitals and seven 4f electrons. In the first calculation, the high spin octet state was solved and, in the second calculation, all 48 sextet states were solved for. These were then mixed under the effect of spin-orbit coupling (SOC) using the well-established spin-orbit restricted active space state interaction (SO-RASSI) approach[76]. The zero-field splitting (ZFS) parameters and the $g$ tensors were extracted from the ab initio results using the SINGLE_ANISO module[77,78].

Relativistically contracted atomic natural orbital basis sets (ANO-RCC) were used in the multireference calculations[79–81]. A valence-polarized quadruple-$\zeta$ basis (VQZP) was used for the Eu ion, valence-polarized triple-$\zeta$ basis sets (VTZP) were used for the N and Si atoms while valence-polarized double-$\zeta$ basis sets (VDZP) were used for the remaining atoms. Cholesky decomposition with a threshold of $10^{-8}$ atomic units was used in the integral storage. Scalar relativistic effects were introduced using the scalar exact two-component (X2C) approach[82–84]. The atomic mean-field integral (AMFI) formalism was used in the construction of the SOC operator used in the SO-RASSI procedure[85,86].

## Data availability

The crystallographic data generated in this study have been deposited in the Cambridge Structural Database under deposition number 2284168. Copies of the data can be obtained free of charge via https://www.ccdc.cam.ac.uk/structures/. The EPR data and ac magnetic susceptibility generated in this study are provided in as Source Data files. Source data for the Supplementary Figs. can be provided upon request. Source data are provided with this paper.

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

## Acknowledgements

D.E., K.L.M.H. and M.M. thank the University of Ottawa, the CFI, and NSERC for financial support of this work. A.M. acknowledges the funding provided by the Academy of Finland (grant no. 332294) and the University of Oulu (Kvantum Institute). Computational resources were provided by CSC-IT Center for Science in Finland and the Finnish Grid and Cloud Infrastructure (persistent identifier urn:nbn:fi:research-infras-2016072533). R.C. and M.R. thank the University of Bordeaux, the Centre National de la Recherche Scientifique (CNRS), the Région Nouvelle Aquitaine, Quantum Matter Bordeaux (QMBx), and the Association Française de Magnétisme Moléculaire. E.V.S. and S.H. acknowledge support from the US Department of Energy, Basic Energy Sciences (DE-SC0020260). Work performed at the National High Magnetic Field Laboratory is supported by the US National Science Foundation (DMR-2128556) and the State of Florida.

## Author contributions

D.E., K.L.M.H. and M.M. conceived the study. D.E. and K.L.M.H. synthesized and characterized the compounds. D.E. and K.L.M.H. collected

X-ray diffraction data on 1, and D.E. performed structure determination and refinement. M.R. and R.C. collected and interpreted the magnetic data. A.M. performed the ab initio calculations and analysis. D.A.G. collected and interpreted the luminescence data. E.V.S, J.v.T. and S.H. collected and interpreted the EPR data. M.M. supervised all aspects of the project. The manuscript was written with contributions from all authors.

## Competing interests

The authors declare no competing interests.
