## [Peer Review File · Nature Communications]

REVIEWER COMMENTS

Reviewer #1 (Remarks to the Author):

The manuscript of D. Errulat et al., communicated by S. Hill, R. Clérac and M. Murugesu represents a piece of novel data about the slow magnetic relaxation in the family of Eu(II) mononuclear complexes. The synthetic part, analysis and molecular spectra along with the X-ray structure analysis are done at the professional level. The conclusion withdrawn from the ESR spectra and DC magnetometry are correct.

A detailed inspection to the manuscript confirms that also the AC magnetic data are interpreted correctly. However, still there are minor questions that could be easily answered by those prominent authors.

1. Is DC magnetometry sensitive enough to distinguish between sign of the small D-parameter?
2. Please correct misprint $D/k_B = -0.65$ ($KD/hc = -0.455 \text{ cm}^{-1}$).
3. Why the data in Fig. S14 were cut at 5 K while those in Fig. 3 were fitted until 8 K?
4. Please construct the plot $\ln(\tau)$ vs $\ln(T)$ until 8 K and follow the curvature at high temperature. If this is linear, then the Raman-like process is on, if non-linear then the Orbach one could prevail.
5. Dozens (hundreds) examples exist where the D-parameter does not reproduce the barrier to spin reversal $U_{\text{eff}} \sim |D|(S^*S - \frac{1}{4})$ assigned to the Orbach relaxation mechanism. Why in the present case is such a nice match?
6. "Nonetheless, a small number of GdIII complexes displaying slow magnetic relaxation under an applied field have been documented.^{21–35}" This number no longer is small.
7. "which is to the best of our knowledge, the first example of a europium SMM." This is not SMM. The compound exhibits a field supported slow magnetic relaxation.

I am recommending the publication of the manuscript after minor corrections.

Reviewer #2 (Remarks to the Author):

Recommendation: This paper represents a significant new contribution and is publishable subject to minor revisions noted. Further review is not needed.

Comments: The article addresses for the first time the single molecule magnet (SMM) behavior of an Eu(II) compound. The syntheses procedure follows previous work by the authors based on the Yb (II) analogue, being able to produce a low-coordinate pseudo-axial complex with high SMM potential and fundamental interest. The key point and the relevance of the reported results is to open new lines of research on divalent lanthanides, and in particular Eu(II) in the field of molecular magnetism. Despite the isotropic character of the $4f^7$ valence state, the highly axial two-coordinate geometry results in an axial crystal field interaction that causes the observed slow magnetic relaxation.

The paper is concise, clear, and well written. The experimental data are well analyzed and presented, with a correct assessment of the magnetic relaxational processes. I suggest the authors consider the minor comments and corrections reported below before article is considered for publication:

- The explanation about the magnetic torque effect on the Eu(II) microcrystals at 25 and 50 K in EPR is not convincing. At those high temperatures the magnetic anisotropy is negligible. Could it be that there is instead a shape anisotropy in the crystals? Has this effect also been observed in magnetic measurements?
- Throughout the paper, the magnetic field is expressed in Tesla units (unit of magnetic flux density B). This is correct for $\mu_0 H$, not for magnetic field H. Please correct the units of H to Oe, or kOe, or include the physical constant of vacuum magnetic permeability, μ_0 , when necessary (Figure S6, S7, S10..).
- Please, include the generalized Debye model equations in supplementary material.
- On page 7, correct the spanning energy in K, $3\text{-}4 \text{ cm}^{-1}$ is equivalent to 4-6 K. Also correct the typo $D/k_B = -0.65 \text{ K}$ (brackets).
- On page 10, apply italics to T in temperature and H in field.

Reviewer #3 (Remarks to the Author):

The main noteworthy result is the discovery of slow magnetic relaxation in a complex of Eu(II), leading to the description of this species as a single-molecule magnet (SMM). The broader significance of the result is that the SMM is based on a non-typical lanthanide in an oxidation state that does not normally feature in studies of SMMs. The common lanthanide is Dy, many examples are known with Tb and Er, but the other lanthanides are much less common in the SMM field, and Eu is - seemingly - unprecedented. The oxidation state +3 is ubiquitous, whereas oxidation state +2 is very rare in lanthanide SMMs. In the case of Eu(II), the electronic structure 'should be' anisotropic (4f7), but the authors find that second-order effects mix into the ground state. The second-order effects are responsible for the observed slow magnetic relaxation, which is novel.

The compound in question is characterized in detail, and the explanation of the spectroscopy and magnetism is convincing. A theoretical study is also used to firm-up the experimental data.

This is a borderline case for NComms since we only have a single example in the current paper. Can we make general conclusions based on one data point? Hard to say. Also, the authors will (presumably) agree that the SMM properties are at the lower end of what is known, with a very small energy barrier of 8 K determined in-field and no proper hysteresis. Is the Eu complex a genuine SMM? The broad, general appeal of the paper is not obvious even if the paper is technically excellent and undoubtedly of interest to those working in lanthanide chemistry and molecular magnetism.

From a causal or mechanistic point of view, what is it about this combination metal, oxidation state and ligands that supports the second-order effects? I think this can only be established reliably with more than one example.

Specific comments:

"Most molecular lanthanide complexes possess strong magnetic anisotropy, which makes them attractive candidates as Single-Molecule Magnets (SMMs)." The first part of that sentence probably is correct, but the second is not. If most lanthanide complexes were attractive SMM candidates, we wouldn't see that 95% of them are based on Dy. Of the paramagnetic lanthanides, half are non-Kramers ions and only SMM candidates if the molecular symmetry requirements are met to ensure a bistable ground state. Strong crystal fields and axial geometries certainly play an important role, but what about spin-phonon coupling? With the latter phenomenon, this is hard to control and largely relies on luck.

Some energy barriers are quoted in kelvin units and some in cm⁻¹.

"The magnetic moment is the highest in the case of a half-filled 4f-shell". Do the authors mean the spin contribution to the moment? The largest measured μ_{eff} in a single-ion complex is in Evans' [Cp₃Ho]- complex.

The term "uncommon valences" should probably be "uncommon oxidation states".

Scheme 1. The zigzag line emanating from K requires some clarification. An attempt to indicate a coordination polymer? I'm not sure that this matters.

A large electropositive Eu(II) center would be expected to interact with the flanking Ph rings in the ligand. This aspect needs some attention so that the true coordination number of Eu can be determined, i.e., is this complex 2-coordinate or not?

The attempted Eu(II) oxidation is interesting. Did the authors try reacting the same oxidant with the K salt of the ligand? Some support for the possibility of a ligand-based oxidation could be obtained in this way.

Defining each of the terms in the two rate equations would help the non-expert readers. In the text below these two equations, the field-dependence of the relaxation time is discussed with H expressed as a mixture of positive and negative powers. Also suggest stating the H terms as written with the equation editor – the exponents are not easy to see.

Reviewer #4 (Remarks to the Author):

In this manuscript, Murugesu et al. reported a quasi-linear bis(silylamido) EuII complex, [EuII(N{SiMePh₂}₂)₂], using a sterically bulky ligand K(THF)N(SiPh₂Me)₂. Its solid structure was determined by single crystal X-ray diffraction. In term of photoluminescence behavior, it exhibited typical emission band of EuII ion from 5d→4f transitions and for magnetic properties, it is a field-induced single-molecule magnet with an effective energy barrier of approximately 8 K. In fact, the first near-linear Eu(II) complex was reported by Mills in 2016 (Inorg. Chem. 2016, 55, 10057) and the photoluminescence properties and mechanisms of Eu(II) compounds were also studied widely. Impressively, some Eu(II) compounds showed high air stability and exciton utilization efficiency around 100 % in organic light-emitting diodes (Angew. Chem. Int. Ed. 2020, 59, 19011). In addition, the synthetic method is very conventional, which is in accordance with the previously reported procedure for [YbII{N(SiMePh₂)₂}₂] (Nat. Chem., 2023, 15, 1100). For magnetic properties, although frequency dependence of the out-of-phase magnetic susceptibility showed peaks in the temperature of 1.6-4.9 K, an additional applied dc field and high frequency (~10000 Hz) ac oscillating field are needed. Therefore, I do not believe that the significance of the work is sufficient for publication in Nat. Commun. as neither luminescent properties nor SMM behaviors of the reported compound provide a new conceptual or a novel insight into the underlying challenge in luminescent coordination compounds or single-molecule magnets. I suggest the publication in a more specialized journal.

There are several issues that need to be addressed.

1. All fitting parameters should be stated with the esd value.
2. The data and discussions of luminescence lifetime and quantum efficiency of the compound should be included.
3. In Introduction, the second sentence should be checked and corrected.
4. The authors should elucidate why the quasi-linear structure doesn't induce excellent SMM properties.
5. The axial anisotropy of the compound is mainly from second-order zero-field splitting, however, such splitting is quite weak, so it is better to give some perspectives about how to preserve the first-order zero-field splitting in Ln(II) compounds.

RESPONSE TO REVIEWERS' COMMENTS

Reviewer #1 (Remarks to the Author):

The manuscript of D. Errulat et al., communicated by S. Hill, R. Clérac and M. Murugesu represents a piece of novel data about the slow magnetic relaxation in the family of Eu(II) mononuclear complexes. The synthetic part, analysis and molecular spectra along with the X-ray structure analysis are done at the professional level. The conclusion withdrawn from the ESR spectra and DC magnetometry are correct.

A detailed inspection to the manuscript confirms that also the AC magnetic data are interpreted correctly. However, still there are minor questions that could be easily answered by those prominent authors.

[...]

I am recommending the publication of the manuscript after minor corrections.

Response: We thank the reviewer for their constructive feedback and positive assessment of our work. Responses to the specific points raised are below.

1. Is DC magnetometry sensitive enough to distinguish between sign of the small D-parameter?

Response: Determination of the sign of the D-parameter using magnetometry can be difficult as data can often be reproduced equally well with both negative and positive values. As a result, it is beneficial to use DC magnetometry in conjunction with other techniques such as EPR, as we have, that can provide complementary information about the D-parameter, including its sign.

2. Please correct misprint $D/k_B = -0.65$ ($KD/hc = -0.455 \text{ cm}^{-1}$).

Response: The text has been corrected.

3. Why the data in Fig. S14 were cut at 5 K while those in Fig. 3 were fitted until 8 K?

Response: The inset of Figure 3 also shows reliable fits up to 5 K to estimate the relaxation time, plotted as the inverse temperature ($T^{-1} = 0.2 \text{ K}^{-1} = 5 \text{ K}$). This temperature range corresponds to the temperatures for which the ac fits are reliable for the characteristic relaxation time accessible with ac frequency up to 10 kHz. In order to alleviate any possible confusion, we have included the temperature range of the inset figure in the caption to further clarify the fitted temperatures. Please note that it is possible to extract in a reliable manner the χ'_0 value up to 8 K.

4. Please construct the plot $\ln(\tau)$ vs $\ln(T)$ until 8 K and follow the curvature at high temperature. If this is linear, then the Raman-like process is on, if non-linear then the Orbach one could prevail.

Response: We don't have any reliable estimation of the relaxation time above 5 K for which the characteristic time is outside our experimental window of ac frequency, so it is not possible to

build a $\ln(\tau)$ vs. $\ln(T)$ plot until 8 K. Nonetheless, please see below the $-\ln(\tau)$ vs. $\ln(T)$ plot until 5 K as unambiguous support for the presence of Orbach relaxation.

5. Dozens (hundreds) examples exist where the D-parameter does not reproduce the barrier to spin reversal $U_{\text{eff}} \sim |D|(S^*S - 1/4)$ assigned to the Orbach relaxation mechanism. Why in the present case is such a nice match?

Response: This is often the case for examples that undergo several competing relaxation mechanisms. In compound **1**, the relaxation dynamics are almost entirely governed by thermally activated Orbach relaxation. As a result, the relaxation will follow the Arrhenius law and so the reproduction is quite accurate and agrees well with theory.

6. “Nonetheless, a small number of Gd^{III} complexes displaying slow magnetic relaxation under an applied field have been documented.^{21–35}” This number no longer is small.

Response: Our use of the term “small” in describing the number of documented Gd^{III} complexes was meant to provide relative context. Considering the vast number of Gd^{III} complexes and the prevalence of slow magnetic relaxation in other lanthanides ions such as Dy^{III}, the instances cited represent a modest proportion in comparison. Nonetheless, to ensure clarity and avoid potential misunderstandings, we have revised the sentence to a more neutral statement.

“Nonetheless, some Gd^{III} complexes displaying slow magnetic relaxation under an applied field have been documented (21–35)”

7. “which is to the best of our knowledge, the first example of a europium SMM.” This is not SMM. The compound exhibits a field supported slow magnetic relaxation.

Response: We understand the concern of the reviewer about the definition of an SMM because there is indeed a lot of confusion on this particular point in the community. It is first important to understand that an SMM has its own intrinsic relaxation time that depends generally on the temperature (Direct, Raman, Orbach-like) and the applied magnetic field (Direct, Raman, Orbach-like, QTM). In order to probe this intrinsic relaxation time, the experimental magnetic set-up (ac or dc) must have a comparable characteristic time. Unfortunately, in many systems like the present one, the relaxation time of the complexes is too fast to be probed by ac techniques (and thus of course by dc techniques) that are usually available up to 10 kHz. Nevertheless, as this relaxation time of an SMM depends often on the applied magnetic field, the application of a magnetic field can allow to change the relaxation time of the SMM such that it falls in the experimental window of measurements. Then the SMM properties of the complex that were not observed in zero-field due to the technical limitations can be observed under a magnetic field. So the SMM properties are **not induced** by the magnetic field but rather **revealed by** the magnetic field in order to have the relaxation time of the SMM in the experimental window (like here between 10 and 10 000 Hz).

So independently of the applied magnetic field, when slow relaxation of the magnetization is observed, the complex should be considered as an SMM. On a related topic, please see also our reply to **Reviewer #3**.

Reviewer #2 (Remarks to the Author):

Recommendation: This paper represents a significant new contribution and is publishable subject to minor revisions noted. Further review is not needed.

Comments: The article addresses for the first time the single molecule magnet (SMM) behavior of an Eu(II) compound. The syntheses procedure follows previous work by the authors based on the Yb (II) analogue, being able to produce a low-coordinate pseudo-axial complex with high SMM potential and fundamental interest. The key point and the relevance of the reported results is to open new lines of research on divalent lanthanides, and in particular Eu(II) in the field of molecular magnetism. Despite the isotropic character of the 4f⁷ valence state, the highly axial two-coordinate geometry results in an axial crystal field interaction that causes the observed slow magnetic relaxation.

The paper is concise, clear, and well written. The experimental data are well analyzed and presented, with a correct assessment of the magnetic relaxational processes. I suggest the authors consider the minor comments and corrections reported below before article is considered for publication:

Response: We sincerely thank the reviewer for the constructive comments and insights regarding our manuscript. We are pleased that the reviewer considers the work of interest to the broad readership of *Nature Communications*. We hope that the following comments have sufficiently addressed the reviewer’s suggestions.

- The explanation about the magnetic torque effect on the Eu(II) microcrystals at 25 and 50 K in EPR is not convincing. At those high temperatures the magnetic anisotropy is negligible. Could it be that there is instead a shape anisotropy in the crystals? Has this effect also been observed in magnetic measurements?

Response: This is a situation that we encounter frequently when performing high-field EPR on unconstrained powders. In the present experiment, the powder was unconstrained because of its reactivity, i.e., the powder was pre-sealed in a quartz tubes. Indeed, it is such a common occurrence that we are writing a technical paper about it, although it will not be submitted until next year. Nevertheless, it is a well understood phenomenon, and we often use it to deliberately align powder samples in order to enhance the parallel components of the EPR spectra (it also demonstrates how careful one should be performing, e.g., magnetic measurements on unconstrained powders). The alignment takes place at the lowest temperature of 5 K. You can even see it in the first few spectra that are recorded, because they are often noisy and variable due to the movement of the crystallites. However, after a few sweeps, the powder settles into a new stable aligned configuration, and all spectra recorded subsequently exhibit the features of the aligned sample, including those recorded at higher temperatures such as 50 K. There are at least two other publications from our group where the alignment of crystallites associated with structures containing two or more differently oriented molecules is observed [Krzystek, J.; Telsler, *J. Dalton Trans. (Perspective)* **2016**, 45, 16751-16763. (Figure 4) and Martínez-Lillo, J. et al., *J. Am. Chem. Soc.* **2013**, 135, 13737-13748 (see Section 3.2.3)]. Unfortunately, those papers do not go into great detail of the alignment (the 2nd one gives a brief discussion), but the analysis proceeds in exactly the same way as the present manuscript. It is because of this particular referee comment and discussions with collaborators that we intend to write a technical paper on this phenomenon in the next months. In the meantime, we cite the 2nd paper above in the EPR section of the revised manuscript.

- Throughout the paper, the magnetic field is expressed in Tesla units (unit of magnetic flux density B). This is correct for $\mu_0 H$, not for magnetic field H. Please correct the units of H to Oe, or kOe, or include the physical constant of vacuum magnetic permeability, μ_0 , when necessary (Figure S6, S7, S10..).

Response: The magnetic field has been changed to kOe in the text and corresponding figures.

- Please, include the generalized Debye model equations in supplementary material.

Response: This has been included in the ESI and has been referenced in the manuscript.

- On page 7, correct the spanning energy in K, 3-4 cm^{-1} is equivalent to 4-6 K. Also correct the typo $D/k_B = -0.65 \text{ K}$ (brackets).

Response: These have been corrected.

- On page 10, apply italics to T in temperature and H in field.

Response: We have italicized the symbols on page 10 and have reviewed the text to ensure that all symbols have been italicized.

Reviewer #3 (Remarks to the Author):

The main noteworthy result is the discovery of slow magnetic relaxation in a complex of Eu(II), leading to the description of this species as a single-molecule magnet (SMM). The broader significance of the result is that the SMM is based on a non-typical lanthanide in an oxidation state that does not normally feature in studies of SMMs. The common lanthanide is Dy, many examples are known with Tb and Er, but the other lanthanides are much less common in the SMM field, and Eu is - seemingly - unprecedented. The oxidation state +3 is ubiquitous, whereas oxidation state +2 is very rare in lanthanide SMMs. In the case of Eu(II), the electronic structure 'should be' anisotropic (4f⁷), but the authors find that second-order effects mix into the ground state. The second-order effects are responsible for the observed slow magnetic relaxation, which is novel.

The compound in question is characterized in detail, and the explanation of the spectroscopy and magnetism is convincing. A theoretical study is also used to firm-up the experimental data.

This is a borderline case for NComms since we only have a single example in the current paper. Can we make general conclusions based on one data point? Hard to say. Also, the authors will (presumably) agree that the SMM properties are at the lower end of what is known, with a very small energy barrier of 8 K determined in-field and no proper hysteresis. Is the Eu complex a genuine SMM? The broad, general appeal of the paper is not obvious even if the paper is technically excellent and undoubtedly of interest to those working in lanthanide chemistry and molecular magnetism.

From a causal or mechanistic point of view, what is it about this combination metal, oxidation state and ligands that supports the second-order effects? I think this can only be established reliably with more than one example.

Response: We greatly appreciate the thorough and constructive feedback provided on our manuscript. The novelty and significance of slow magnetic relaxation in a Eu(II) complex have been rightly emphasized. However, we wish to further emphasize that our submission is intended as a communication article, which focuses on sharing what we believe is a timely, significant, and novel example of molecular magnetism in a Eu^{II} complex. While we recognize the inherent limitations of a single example, we believe that these results warrant rapid broadcast to the broader research community and are of immediate interest to specialists and broad audiences alike. The journal's aim & scope indicate that papers published by *Nature Communications* aim to represent important advances of significance to specialists within each field, and we believe that this work wholeheartedly falls within this category. Our hope is that this communication article will motivate

researchers to explore similar systems and spur investigations into the SMM properties of divalent europium complexes.

Sufficiently slow magnetic relaxation manifests as a hysteresis loop when the magnetic moment is slowed enough that it is prevented from relaxing to equilibrium (in our available experimental window). The occurrence of slow magnetic relaxation and magnetic blocking are therefore intimately linked and both properties ultimately result from the same magnetic bistability and magnetic anisotropy that give rise to a quantized energy barrier to spin reversal, which is undoubtedly present in **1**. While we recognize the importance of blocking temperature as a figure of merit, there are many cases of SMMs where large energy barriers do not necessarily translate into large blocking temperatures, even at temperatures of 2 K (such as Chilton *et al.*, *Dalton Trans.* **2019**, 48, 10795-10798).

On a related topic, please see also our reply to point 7 of **Reviewer #1**.

Specific comments:

“Most molecular lanthanide complexes possess strong magnetic anisotropy, which makes them attractive candidates as Single-Molecule Magnets (SMMs).” The first part of that sentence probably is correct, but the second is not. If most lanthanide complexes were attractive SMM candidates, we wouldn't see that 95% of them are based on Dy. Of the paramagnetic lanthanides, half are non-Kramers ions and only SMM candidates if the molecular symmetry requirements are met to ensure a bistable ground state. Strong crystal fields and axial geometries certainly play an important role, but what about spin-phonon coupling? With the latter phenomenon, this is hard to control and largely relies on luck.

Response: Following the advice of the reviewers, the beginning of the introduction was eliminated as the rest of the introduction better describes the properties that give rise to an attractive Ln-based SMM.

Some energy barriers are quoted in kelvin units and some in cm^{-1} .

Response: For the sake of consistency, the energy barriers have all been quoted in Kelvin. Acknowledging the prevalence of both conventions, the energy is also presented as cm^{-1} in parenthesis.

“The magnetic moment is the highest in the case of a half-filled 4f-shell”. Do the authors mean the spin contribution to the moment? The largest measured μ_{eff} in a single-ion complex is in Evans' [Cp₃Ho]- complex.

Response: We thank the reviewer for the suggestion, this was our intended meaning and the sentence has been modified to provide clarity regarding the spin contribution.

The term “uncommon valences” should probably be “uncommon oxidation states”.

Response: We have made this change.

Scheme 1. The zigzag line emanating from K requires some clarification. An attempt to indicate a coordination polymer? I'm not sure that this matters.

Response: The reviewer is correct that this was included to indicate an extended network. We have removed it to avoid distractions from the synthetic scheme.

A large electropositive Eu(II) center would be expected to interact with the flanking Ph rings in the ligand. This aspect needs some attention so that the true coordination number of Eu can be determined, i.e., is this complex 2-coordinate or not?

Response: These interactions are highlighted in the structural description along with distances of the relevant Eu•••C interactions that occur with flanking phenyl rings. These close contacts are well known to occur in lanthanides and are in line with those of previous reports of Ln^{II} complexes of this sort (*Nat. Chem.*, **2023**, 15, 1100; *Inorg. Chem.*, **2016**, 55, 10057). As a result, these *bis*(silylamido) complexes have generally come to be referred to by their formal two-coordinate nature, a category compound **1** is part of (see *Dalton Trans.*, **2020**, 49, 14320-14337; *ACS Omega*, **2018**, 3, 9462–9475). Bar *et al.* in *Coord. Chem. Rev.*, **2018**, 367, 163-216 also provides a commentary regarding the commonplace definition of coordination number in their section covering formally two-coordinate lanthanide complexes.

Nonetheless, we have modified some of the text to not invoke this commonplace classification in order to avoid any possible confusion.

The attempted Eu(II) oxidation is interesting. Did the authors try reacting the same oxidant with the K salt of the ligand? Some support for the possibility of a ligand-based oxidation could be obtained in this way

Response: This was attempted as confirmation of the ligand-centred oxidation. This reaction similarly resulted in the deposition of a Ag(0) mirror and the formation of an intractable product whose structure could not be determined.

Given the lack of clear reaction products, we had originally opted to not include this remark. However, it is now clear to us that this would be of interest to the reader. As such, a remark was added to the discussion about this as further evidence of ligand-centred oxidation.

Defining each of the terms in the two rate equations would help the non-expert readers. In the text below these two equations, the field-dependence of the relaxation time is discussed with H expressed as a mixture of positive and negative powers. Also suggest stating the H terms as written with the equation editor – the exponents are not easy to see.

Response: We have included the terms in the discussion of the rate equations and have rewritten the H dependence in the equation editor. We hope the changes provide greater context to the reader and alleviate any confusion and clarity issues.

In regards to the field dependence, the sign of the power is of importance and relates to what kind of dependence the relaxation process has on the magnetic field. For example, in Eqn 1., the relaxation time of QTM increases with a square dependence on the magnetic field (H^2), while the relaxation times of the Direct relaxation decreases with field (H^{-4}).

Reviewer #4 (Remarks to the Author):

In this manuscript, Murugesu et al. reported a quasi-linear bis(silylamido) EuII complex, $[\text{EuII}(\text{N}\{\text{SiMePh}_2\}_2)_2]$, using a sterically bulky ligand $\text{K}(\text{THF})\text{N}(\text{SiPh}_2\text{Me})_2$. Its solid structure was determined by single crystal X-ray diffraction. In term of photoluminescence behavior, it exhibited typical emission band of EuII ion from $5d \rightarrow 4f$ transitions and for magnetic properties, it is a field-induced single-molecule magnet with an effective energy barrier of approximately 8 K. In fact, the first near-linear Eu(II) complex was reported by Mills in 2016 (*Inorg. Chem.* 2016, 55, 10057) and the photoluminescence properties and mechanisms of Eu(II) compounds were also studied widely. Impressively, some Eu(II) compounds showed high air stability and exciton utilization efficiency around 100 % in organic light-emitting diodes (*Angew. Chem. Int. Ed.* 2020, 59, 19011). In addition, the synthetic method is very conventional, which is in accordance with the previously reported procedure for $[\text{YbII}\{\text{N}(\text{SiMePh}_2)_2\}_2]$ (*Nat. Chem.*, 2023, 15, 1100). For magnetic properties, although frequency dependence of the out-of-phase magnetic susceptibility showed peaks in the temperature of 1.6-4.9 K, an additional applied dc field and high frequency (~ 10000 Hz) ac oscillating field are needed. Therefore, I do not believe that the significance of the work is sufficient for publication in *Nat. Commun.* as neither luminescent properties nor SMM behaviors of the reported compound provide a new conceptual or a novel insight into the underlying challenge in luminescent coordination compounds or single-molecule magnets. I suggest the publication in a more specialized journal.

Response: There is yet to be a description of the slow relaxation of the magnetization in a Eu^{II} complex; this phenomenon and associated measurements were also notably absent in the earlier work by Mills *et al.* (*Inorg. Chem.* 2016, 55, 10057) that the reviewer pointed out. Given that an Eu complex in this context is unprecedented, we fear that the reviewer may have failed to place the manuscript within the appropriate context.

Moreover, it should also be noted that the reference the reviewer is mentioning (*Inorg. Chem.*, 2016, 55, 10057) presents an incorrect interpretation of the luminescence of the divalent europium complex reported there. The authors are deconvoluting the bands and assigning the transition to states from trivalent europium rather than a d-f transition. The incorrect assignments can be seen in the sentence:

“The resultant spectrum exhibits two broad peaks centered at 645 (15500) and 600 (16670) nm (cm^{-1}) and a less intense sharper peak at 559 nm (17890 cm^{-1}), which are most likely due to f-f transitions corresponding to $^5D_0 \rightarrow ^7F_3$, $^5D_0 \rightarrow ^7F_2$ or 7F_1 , and $^5D_0 \rightarrow ^7F_0$, respectively.”

There are several issues that need to be addressed.

1. All fitting parameters should be stated with the esd value.

Response: The esd for the fitting parameters have been included.

2. The data and discussions of luminescence lifetime and quantum efficiency of the compound should be included.

Response: We are now including the emission decay (luminescence lifetime) for **1** at room temperature. The obtained value is 2.45 μ s. The plot has been provided as Figure S4 and a brief discussion has been included in the manuscript.

Regarding quantum efficiency, the reviewer may be mistaken about the term. Discussions about exciton utilization or quantum efficiency are more appropriate when describing LED devices or solar cells. We believe the reviewer is referring to quantum yield. Unfortunately, we are not able to measure the quantum yield of air-sensitive samples with our experimental setup. It should be noted that the luminescence properties of **1** are not the main focus of the manuscript, we are mainly using the emission to probe the divalent state and the SMM properties. Our commitment to this primary focus is underscored by the results we've highlighted and the discussions that form the backbone of our manuscript.

3. In Introduction, the second sentence should be checked and corrected.

Response: The introduction was modified in accordance with the suggestions from the reviewers.

4. The authors should elucidate why the quasi-linear structure doesn't induce excellent SMM properties.

Response: As described in the manuscript, the spherically symmetric charge density of the $^8S_{7/2}$ ground state lacks orbital angular momentum. As such, the magnetic properties are only a result of second order SOC because there is no first order orbital moment. This description is the basis for the origin of slow magnetization relaxation.

5. The axial anisotropy of the compound is mainly from second-order zero-field splitting, however, such splitting is quite weak, so it is better to give some perspectives about how to preserve the first-order zero-field splitting in Ln(II) compounds.

Response: In the context of this report, it is challenging to offer a perspective on first-order zero-field splitting as this is not present in **1**. Ultimately, you would have to go to a different lanthanide (such as Dy^{III}) to achieve this.

REVIEWER COMMENTS

Reviewer #1 (Remarks to the Author):

After acceptance of the reviewer's comments and giving additional explanations, the manuscript is now ready for publication as it is.

Reviewer #2 (Remarks to the Author):

I have read the authors' responses, and find that the manuscript has been carefully revised. I consider that all concerns raised by myself and other referees have been fully addressed, so I would be glad to recommend for acceptance as is.

Reviewer #3 (Remarks to the Author):

Revised manuscript is fine.
Authors have responded positively to points raised by all reviewers.
Recommend accept.

Reviewer #4 (Remarks to the Author):

After reading carefully the reply to the reviewer comments from the authors, the manuscript and SI material, the referee has not changed mind when it comes to the scientific novelty of this work. Although it is competently done, the referee doesn't see in the reported results what should justify this work to be published in a broad audience journal of the caliber of Nature Communications: 1) By all the traditional metrics of SMMs, the Eu(II) compound is at best a field-induced SMM (as also mentioned by Reviewers 1 and 3). In the vast majority of cases, the maximum ac frequency used to probe slow magnetic relaxation is 1000 Hz. Here, many relaxation peaks of out-of-phase ac magnetic susceptibility can only be observed at frequency above 2000 Hz and under a dc magnetic field of 800 Oe, indicating that it has very weak magnetic anisotropy. It's well-known that Eu(III) and Eu(II) ions are not ideal candidates as the former gives $J = 0$ and the latter is magnetically isotropic. As a result, we don't think this work will "motivate researchers to explore similar systems and spur investigations into the SMM properties of divalent europium complexes"; 2) From synthetic point of view, it is a common salt metathesis reaction of a bulky ligand and EuI₂ (commercially available) where the ligand has been reported in another isostructural Yb(II) compound; 3) The fitting results in Figure 3 are not convincing. For this case, the authors should give all possible relaxation mechanisms and fitting results, then comparing them and select the best fit (see Supplementary Fig. 8 in Nat. Commun. 2022, 13, 2014). All fitting results should give the value of R²; 4) A detailed explanation is necessary for the abnormal value of τ_0 . In addition, all computational details should be provided in Supporting Information.

RESPONSE TO REVIEWERS' COMMENTS

Reviewer #1 (Remarks to the Author):

After acceptance of the reviewer's comments and giving additional explanations, the manuscript is now ready for publication as it is.

We thank the reviewer for taking their time reviewing our manuscript and providing us with critical constructive feedback in the revision process.

Reviewer #2 (Remarks to the Author):

I have read the authors' responses, and find that the manuscript has been carefully revised. I consider that all concerns raised by myself and other referees have been fully addressed, so I would be glad to recommend for acceptance as is.

We thank the reviewer for taking their time reviewing our manuscript and providing us with critical constructive feedback in the revision process.

Reviewer #3 (Remarks to the Author):

Revised manuscript is fine. Authors have responded positively to points raised by all reviewers. Recommend accept.

We thank the reviewer for taking their time reviewing our manuscript and providing us with critical constructive feedback in the revision process.

Reviewer #4 (Remarks to the Author):

After reading carefully the reply to the reviewer comments from the authors, the manuscript and SI material, the referee has not changed mind when it comes to the scientific novelty of this work. Although it is competently done, the referee doesn't see in the reported results what should justify this work to be published in a broad audience journal of the caliber of *Nature Communications*.

However, we respectfully hold a differing opinion regarding their assessment of novelty. This manuscript showcases the impact of a meticulous ligand design coupled with an optimal crystal field (especially in the context of a near-linear system) which leads to SMM behaviour even in scenarios **where it is unexpected**. Consequently, we firmly believe that this work offers fresh perspectives within the realms of SMM, magnetism, and material science. Such contributions warrant publication in *Nature Communications*. Moreover, this viewpoint is reinforced by the unanimous support of the **other three reviewers** who have advocated for the manuscript's publication in its current form.

1) By all the traditional metrics of SMMs, the Eu(II) compound is at best a field-induced SMM (as also mentioned by Reviewers 1 and 3). In the vast majority of cases, the maximum ac frequency used to probe slow magnetic relaxation is 1000 Hz. Here, many relaxation peaks of out-of-phase ac magnetic susceptibility can only be observed at frequency above 2000 Hz and under a dc magnetic field of 800 Oe, indicating that it has very weak magnetic anisotropy.

We acknowledge that the term "field-induced SMM" has been conventionally employed within the molecular magnetism community. Nevertheless, it's important to highlight that **this term is not accurate**, and the scientific community is gradually moving away from its usage. As previously emphasized in our response, it's crucial to note that the applied static field **does not induce the SMM behavior, but rather revealed by the magnetic field in order to have the relaxation time of the SMM in the experimental window (like here between 10 and 10 000 Hz)**.

In most research laboratories, the maximum ac frequency employed to investigate slow magnetic relaxation typically caps at 1000 Hz due to equipment limitations. However, in our case, with our instrument capabilities we can expand our experimental range to 10 kHz. This extended frequency range allows for a more detailed and comprehensive study of our Eu complex. This extended experimental window enables us to observe and thoroughly investigate these dynamics, as evidenced by our current study.

It's well-known that Eu(III) and Eu(II) ions are not ideal candidates as the former gives $J = 0$ and the latter is magnetically isotropic. As a result, we don't think this work will "motivate researchers to explore similar systems and spur investigations into the SMM properties of divalent europium complexes"

Respectfully, we disagree with the reviewer's viewpoint, which seems somewhat narrow. It's crucial to note that the lower oxidation state of Eu(II) renders it more prone to forming stronger exchange interactions. Consequently, the Eu(II) ion holds potential for application in polymetallic systems that could exhibit intriguing properties. Presently, there is a significant emphasis on comprehending and enhancing exchange interactions within 4f elements while preserving magnetic anisotropy. Additionally, the challenge of metal-ligand covalency in 4f elements persists, necessitating an understanding fostered by unique ligands, unconventional coordination environments (such as 2-coordinate systems), and distinct geometries. Thus, Eu(II) holds complexities that extend beyond a mere opinion suggesting it's "not a good candidate for SMMs!"

2) From synthetic point of view, it is a common salt metathesis reaction of a bulky ligand and EuI₂ (commercially available) where the ligand has been reported in another isostructural Yb(II) compound;

While the synthetic approach might appear as a straightforward metathesis reaction, executing it is far from trivial. Particularly, the crystallization techniques and conditions necessary to obtain single crystals suitable for X-ray and other analyses present significant challenges. Despite the prediction of "linear" two-coordinate systems as potential candidates for high-performance systems, there are scant examples in the literature, with only one currently showcasing SMM behavior published in *Nature Chemistry* 2023.

If this were a common metathesis reaction devoid of a cleverly designed system, numerous reports would have emerged by now, which is not the case. Hence, **we respectfully disagree with the referee's comments that appear to undermine and oversimplify the targeted design and execution of this unique Eu(II) SMM.**

3) The fitting results in Figure 3 are not convincing. For this case, the authors should give all possible relaxation mechanisms and fitting results, then comparing them and select the best fit (see Supplementary Fig. 8 in Nat. Commun. 2022, 13, 2014). All fitting results should give the value of R²;

We would like to kindly redirect the reviewer to main text and carefully analyse our fits. In the main body of the text, a comprehensive elucidation of the correlation between the relaxation time temperature (referenced in Figure 3 and Figure S15) and field dependencies (shown in Figure S11) is thoroughly provided. Within our experimental window, there appears to be an absence of direct relaxation concerning the field dependence beyond 1 kOe. However, the phenomenon of Quantum Tunneling of Magnetization (QTM) manifests at lower fields, specifically below 1 kOe. The analysis of temperature dependence involves the consideration of both QTM (which remains constant across temperatures) and a relaxation akin to the Orbach process. These elements are integrated into the fitting procedure, resulting in the derivation of only two parameters (Δ and τ_0) that satisfactorily align with a linear trend. Remarkably, the regression factor (R) for this linear regression fit is explicitly presented in the text as $R = 0.99991$! This demonstrates a commendable level of regression. Alternatively, if the referee prefers, the coefficient of determination (R^2) stands at 0.9998, indicating an equally robust fit. The quality of these fits reflects the current state-of-the-art fitting procedures, offering a high level of accuracy. We are perplexed as to why the referee has deemed them unconvincing, especially when three other reviewers have concurred with our fitting approach and assessment.

4) A detailed explanation is necessary for the abnormal value of τ_0 .

We would like to refer the reviewer to the main text of the manuscript. In this section, it is explicitly mentioned that the significant "pre-exponential factor (τ_0) of $3.1(6) \times 10^{-6}$ s" might potentially stem from an additional relaxation mechanism. However, determining this mechanism becomes challenging due to overparametrization of the data, which occurs when attempting to fit a straight line with more than two parameters. The passage reads:

"Nonetheless, two factors suggest the presence of additional relaxation mechanisms: (i) the obtained pre-exponential factor (τ_0) of $3.1(6) \times 10^{-6}$ s is larger than expected for an Orbach-like relaxation regime; and (ii) the field dependence of the relaxation time above 1 kOe is complex and could not be reproduced by fitting to Equation 2. Attempts to fit the experimental data with relaxation models including Raman and direct processes lead to overparameterization and did not improve the theory-experiment agreement above 0.1 T (Figure S15)."

-In addition, all computational details should be provided in Supporting Information.

With all due respect, we are bit concerned that the reviewer did not carefully read the manuscript, as the computational details are given in full detail in the methods section of the manuscript as required for *a*. The methods are described in much more detail than in a typical SMM paper ensuring the thorough analysis of our reported compound.